# The Importance of Groundwater Quality and Other Habitat Parameters for Effective Active Protection of an Endangered Plant Species in Eastern Poland

Artur Serafin [1], Magdalena Pogorzelec [2,*], Barbara Banach-Albińska [3], Ewa Zalewska [4], Urszula Bronowicka-Mielniczuk [5] and Michał Arciszewski [2]

1  Department of Environmental Engineering and Geodesy, University of Life Sciences in Lublin, Leszczyńskiego 7, 20-069 Lublin, Poland; artur.serafin@up.lublin.pl
2  Department of Hydrobiology and Protection of Ecosystems, University of Life Sciences in Lublin, Dobrzańskiego 37, 20-262 Lublin, Poland; arciszewski.foto@gmail.com
3  Department of Invertebrate Ecophysiology and Experimental Biology, University of Life Sciences in Lublin, Doświadczalna 50, 20-280 Lublin, Poland; barbara.banach@up.lublin.pl
4  Department of Vegetable and Herb Crops, University of Life Sciences in Lublin, Akademicka 15, 20-950 Lublin, Poland; ewa.zalewska@up.lublin.pl
5  Department of Applied Mathematics and Computer Science, University of Life Sciences in Lublin, Głęboka 28, 20-612 Lublin, Poland; urszula.bronowicka@up.lublin.pl
*  Correspondence: magdalena.pogorzelec@up.lublin.pl

**Abstract:** Anthropogenic habitat transformations involving changes in hydrologic conditions in the peatlands of eastern Poland contribute to the disappearance of sites of numerous relict plant species. The study aimed to verify whether sites chosen for the reintroduction of the endangered species *Salix lapponum* had been well selected by analyzing selected habitat parameters and determining whether changes in their values may in the long term have a significant impact on the functioning of new populations of the species. The results obtained at sites where the *S. lapponum* population was replenished with new individuals were analyzed in relation to data from one of the natural sites of the species. Hydrochemical characterization of the groundwater at the study sites confirmed that there was no significant influx of nutrients into the habitat or other hydrological disturbances due to human activity. The values obtained for the factors tested were within the limits of the specific preferences of the species. Changes in the values of some physical-chemical parameters of the water were due to the ecosystem's internal metabolism, and the site with the highest hydrochemical stability was the one where *S. lapponum* occurred naturally. The species composition and structure of the phytocoenoses at all study sites were characteristic of natural sites of the species and showed no disturbances caused by human impact. The microbiological analyses indicated normal soil processes. The hydrochemical and biocoenotic stability of the habitat, including the microbiological balance of the soil, which was free of pathogenic fungi, should have a positive effect on the condition of the reintroduced plants.

**Keywords:** hydrochemical condition; *Salix lapponum*; reintroduction; habitat condition; fungi; phytocoenoses

## 1. Introduction

The dynamics and effects of global (climatic) and local (anthropogenic) transformations of habitat conditions are disproportionate to the adaptability of boreal plant relicts, despite the fact that the anatomical structure and physiology of species are adapted to life in the often-extreme conditions of wetland ecosystems [1–4]. These species have specific habitat preferences which are usually determined by a narrow range of ecological valence [5]. Their growth and development are affected by even slight changes in habitat conditions, including changes in the physical-chemical properties of shallow groundwater-related, for example, to the agricultural use of the catchment area. Recent studies show that the

streams and concentrations of DOC and nitrates in groundwater are the key outputs of the agricultural impact. However, they differ in terms of time and space due to differences in hydrology, physical and chemical properties of the soil, and the manner and intensity of its agricultural use [6,7]. Therefore, it is necessary to understand the physical, biological, and chemical relationships that determine the fate of any organic pollutant in the rhizosphere of the biocenotic complex [8].

The quality of natural surface water and groundwater directly affects individual plants, but also indirectly affects entire biocoenoses, transforming their qualitative and quantitative structure [4,9,10].

Disturbances of the hydrological regime, changes in the biogenic accumulation rate, fluctuations of the groundwater table, and other phenomena caused by human impact lead to changes in the habitat characteristics of peatlands in the central part of eastern Poland—Polesie Podlaskie. They result in the degradation and fragmentation of habitats which are often the last refuges of many rare plant species, including boreal relics [3,4,11–14].

Knowledge of the mechanisms and effects of changes taking place in natural habitats is of fundamental importance in planning strategies and methods of active conservation of endangered species, including reintroduction. The effectiveness of these efforts depends primarily on the preservation of the balance in the ecological system to which the species is introduced, i.e., the stability of the hydrological and biocenotic systems of the habitat. Extensive management inhibiting the succession of thicket and forest vegetation, applied as needed following the properly executed reintroduction of the species, may also prove to be extremely important [3,4,11].

Selection of optimal habitats for the reintroduction or replenishment of populations requires knowledge of the species' preferences in terms of key abiotic and biocenotic habitat features and current and anticipated climate characteristics [11,15,16].

An example of a relict species for which reintroduction was chosen as the optimal means of conservation in eastern Poland is *Salix lapponum* L. (downy willow). This relatively short shrub (up to 1 m high), with characteristic silver leaves covered in thick down, is found in high numbers in wetland ecosystems of northern Europe and Asia. It prefers sites on soil that is organogenic, oligotrophic, or mesotrophic, but rich in organic matter, with a neutral, acidic, or moderately acidic reaction [17]. In Eastern and Central Europe, this species is found on the southern limit of its geographic range or in refuges separate from its main range, where it is recognized as a boreal relict. One of the main reasons for the retreat of this species from its natural habitats in Poland is changes in the hydrographic conditions in the habitats. The survival of the species is not favored by fluctuations disturbing the typical seasonal climate characteristics, observed since meteorological measurements have been conducted in Europe, i.e., since the mid-18th century [18]. In Poland, *S. lapponum* is under legal species protection and requires active conservation [4,19,20]. To select optimal habitats for the reintroduction or replenishment of the small existing populations of this species, the rate and degree of their transformation must be considered and, importantly, predicted [21]. The success of active conservation of both this species and other species subjected to this type of intervention largely depends on the appropriate choice of habitat in terms of hydrochemical state and biocenotic structure.

In 2017–2020 the first attempt was made to reintroduce the Pleistocene boreal relict *S. lapponum* in the peatlands of eastern Poland. The micropropagation method was used to obtain more than 6000 individuals for colonization of peatland ecosystems where the species had completely died out or to replenish the small existing populations remaining in isolated sites [22].

Based on field reconnaissance supported by laboratory analyses of the shallow groundwater of peatlands, as well as other multi-faceted ecological analyses, optimal locations for new populations of *S. lapponum* were selected. These were semi-natural habitats with potentially limited human impact and regulated hydrological conditions. One of the criteria for their selection was their situation within the limits of protected natural areas [11].

In 2018–2019 *S. lapponum* individuals were planted on selected sites located in the peatland complexes of Blizionki, Spławy and Durne Bagno and on peatlands near Lakes Długie, Moszne and Karaśne, situated in the Łęczna-Włodawa Lakeland (Pojezierze Łęczyńsko-Włodawskie), in eastern Poland (unpublished data, Figure 1).

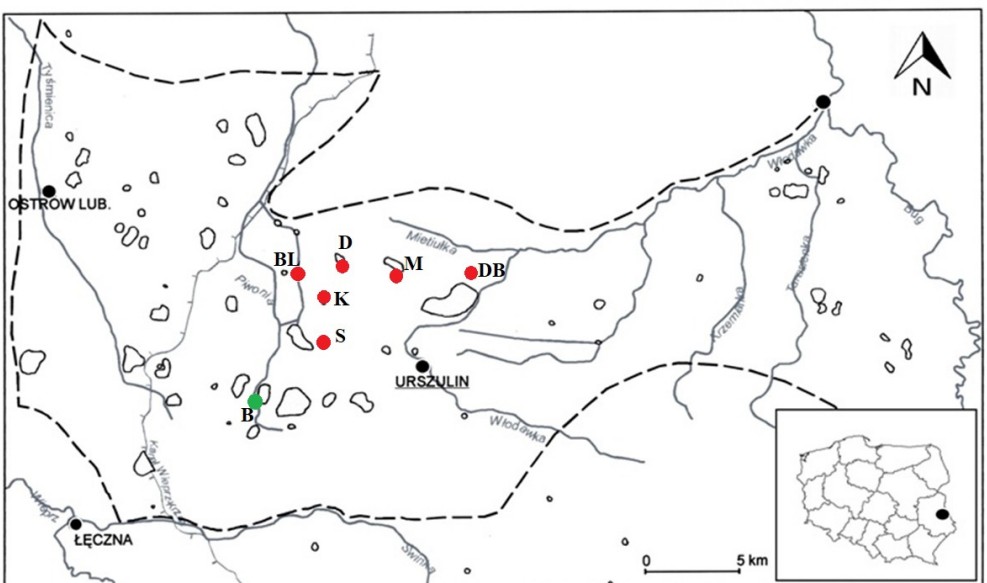

**Figure 1.** Location of sites where *S. lapponum* populations were reintroduced or replenished in eastern Poland (red dots). BL—Blizionki; D—Długie; M—Moszne; DB—Durne Bagno; K—Karaśne; S—Spławy and B—natural site of *S. lapponum* populations (green dot).

Among the many factors potentially influencing the effectiveness of reintroduction, relatively stable, optimal habitat conditions for the species are regarded as crucial. Therefore, habitats must continue to be monitored after reintroduction as well. This makes it possible to confirm that the location for the new populations was well chosen, as well as to predict potential changes in the abiotic environment and biocoenoses, which can unquestionably affect the growth and condition of plants [23–25]. For this reason, hydrochemical, botanical, and microbiological soil analyses were carried out at the sites of the new *S. lapponum* populations, and the results were compared with those obtained during monitoring of the habitat at a site of natural and stable occurrence of *S. lapponum* in eastern Poland. An attempt was made to predict the survival of the population based on data on the state of and changes in selected abiotic factors at the reintroduction sites.

The activities are undertaken to verify the correctness of the selection of sites for the reintroduction process of Salix lapponum based on hydrochemical, botanical, and microbiological analyzes of the soil and to compare the values of selected parameters with the values at the sites of the natural and constant occurrence of the *S. lapponum* population. An additional aim of the study was to determine whether the values of the analyzed parameters during the study period maintain the ranges conducive to the survival of new populations of the reintroduced species in the longer term. The presented research was used to evaluate the activities carried out for the reintroduction of the plant boreal relic.

## 2. Materials and Methods

Based on analysis of historical source materials [26,27], the hydrochemical characteristics of the habitats and data characterizing the biocoenoses, two locations where replenishment of *S. lapponum* populations was carried out were selected for research: the lake and peatland complexes of Moszne (M) and Długie (D) in Polesie National Park. For comparison, we selected a peatland near Lake Bikcze (B, in the buffer zone of Polesie National Park), where there is a natural population of over 300 downy willow individuals [12,26].

Analyses of habitat conditions were carried out in accordance with the procedure of hierarchization of abiotic environmental properties—a methodical study for research on the ecology of plants in peatland habitats [28]. The monitoring procedure included laboratory analyses of the physical-chemical parameters of the shallow groundwater, analysis of the species composition of phytocoenoses, and microbiological analysis of the soil in the rhizosphere of the plants introduced to the ecosystem.

It should be added that the methodology of sampling shallow peat bog waters was related to the specific geological structure of the region, which is quite complicated but has a large impact on the natural conditions.

A series of Mesozoic marine sediments (limestones, marls, Jurassic, and Cretaceous rocks) lying on the Palaeozoic formations are of particular importance here. The highest link in this series is the Upper Cretaceous rocks of coal formed in the form of soft marls, limestones, and writer's chalk. They appear on the surface in the form of low, delicate humps. Tertiary and Quaternary sediments were deposited on the Cretaceous substrate. These are mainly clays and gravels with sands of moraines of the Central Polish Glaciation and sediments of water, river, and lake accumulation: sands, silt, gyttja, and peat layers on raised and transitional fens [29]. For this reason, hydrated peat layers do not come into contact with deeper-fed aquifers and are only fed by the atmosphere, surface runoff, and subsurface infiltration. During 2016–2017 (period I—before reintroduction of *S. lapponum*) and 2018–2019 (period II—after planting at the new sites), 6 times a year (from April to September), groundwater was sampled for laboratory analysis. Water was sampled from soil piezometers (perforated PCV tube, 1 m length and 10 cm diameter with a plug) previously installed at the central part of sites on the peatlands near Lakes Bikcze (B), Długie (D) and Moszne (M). Piezometers were buried in a layer of peat saturated with water (acrotelm) to a depth of 90 cm. Water samples were taken from the piezometer using a weight-loaded container. First, at each stand, all available water was taken from the piezometer, and then, after filling it, the water was immediately taken in the volume of 1.5 dm$^3$ for further laboratory analysis. Laboratory analyses of the qualitative parameters of the shallow groundwater included determination of the following physical-chemical parameters: total nitrogen content $N_{tot}$, the content of nitrites $N\text{-}NO_3$ and nitrates $N\text{-}NO_2$, the content of ammonium nitrogen $N\text{-}NH_4$ (Photometric detection of generated nitrate after UV and thermos digestion via reduction to nitrite and azo dye formation. Flow analysis method according to DIN EN ISO 29441), the content of nitrites $N\text{-}NO_3$ and nitrates $N\text{-}NO_2$ (Photometric via azo dye formation. Flow analysis method according to DIN EN ISO 13395), the content of ammonium nitrogen N-NH4 (Photometric by gas diffusion and color indicator. Flow analysis method according to ISO 11732), the concentration of total phosphorus $P_{tot}$ and phosphates $P\text{-}PO_4$ by spectrophotometry with ammonium molybdate, and dissolved organic carbon content (DOC) using a PASTEL UV automatic analyzer. In addition, the reaction (pH) and electrical conductivity ($\mu S\ cm^{-1}$) of the water samples were determined using a YSI 556 MPS multiparametric probe (YSI, Yellow Springs, OH, USA).

At the selected sites of new *S. lapponum* populations (D and M) and the natural site of the species (B), inventories of flora species were made on representative plots (100–400 m$^2$), one for each study period. Species were identified using Rutkowski's key [30], and the botanical nomenclature was adopted after Mirek et al. [30]. Jaccard's species similarity index was calculated [31], and preliminary syntaxonomic analyses of the phytocoenoses were conducted, i.e., classification of the species to alliances and classes [19].

Microbiological analysis of the soil at the study sites (D and M) was performed in May and June 2019. These analyses included the determination of the number of colonies forming units of bacteria and fungi and the species composition of fungi present in the natural environment. For this purpose, rhizosphere soil, i.e., soil adhering directly to the roots of *S. lapponum* individuals, and soil from outside the rhizosphere were sampled. Microbiological analysis (total numbers of bacteria and fungi) was performed by the serial dilution plate method [32]. The total bacterial count per g DW of soil was determined in soil solutions at dilutions of $10^{-5}$, $10^{-6}$, and $10^{-7}$, on nutrient agar. The total fungal count

in each soil sample was determined in Thayer–Martin agar [33], using $10^{-2}$, $10^{-3}$, and $10^{-4}$ dilutions of the soil solution. Quantitative readings were taken after the appropriate incubation period for each group of microbes: bacteria after 24 or 48 h at 28 °C, and fungi after 2–4 days, also at 28 °C. The number of colony-forming units (CFU) that grew on the solid microbiological media in Petri dishes was determined. Following qualitative analysis to determine the taxonomy of the fungi (to genus or species), the colonies were inoculated on potato dextrose agar (PDA—ready-made product from Difco).

Fungi were identified on standard media using multiple keys for systematic identification. The names of the fungal species obtained were given according to applicable principles of taxonomy, based on the Index Fungorum database [34].

## 3. Results

### 3.1. Hydrochemical Conditions of the Habitats

In the first stage of the research, the empirical distribution of the values of physical-chemical parameters for pooled data from all study sites was analyzed. Variations in the values of these factors determine the ecological tolerance range of the *S. lapponum* individuals for these parameters and the locations of its occurrence (Figures 2 and 3).

In both monitoring periods (I and II) the average values of most of the factors e.g., for the natural site (B) and the sites (D and M) where new individuals replenished the *S. lapponum* population were beyond the range of the typical distribution of observations, specified by the median due to the substantial deviation of some of the results (Figures 2 and 3).

In the case of the site near Lake Bikcze (B), there were only minor differences in the average values of most of the parameters between the two study periods (I and II). Higher means for DOC, $P_{tot,}$ and N-NO$_3$ were noted at this site in the second study period: e.g., pH = 5.5–5.57; N$_{tot.}$ = 1.834–1.695 mg·dm$^{-3}$; P-PO$_4$ = 0.143–0.146 mg·dm$^{-3}$; EC = 149.5–147.9 μS·cm$^{-1}$ (Figures 2 and 3). In contrast, at the sites of reintroduction of downy willow—M and D, the differences were more evident and were noted for: EC$_M$ = 69.0–61.2 μS·cm$^{-1}$; EC$_D$ = 167.125–113.5 μS·cm$^{-1}$; N$_{tot.M}$ = 2.327–2.118 mg·dm$^{-3}$; N$_{tot.D}$ = 3.066–2.295 mg·dm$^{-3}$; N-NO$_{3M}$ = 0.249–0.046 mg·dm$^{-3}$; N-NO$_{3D}$ = 0.2–0.015 mg·dm$^{-3}$ (lower average in period II) and DOC$_M$ = 21.39–33.35 mg·dm$^{-3}$; DOC$_D$ = 24.94–56.32 (higher average in period II) (Figures 2 and 3).

Factors showing relatively minor variation, both at the natural site of *S. lapponum* (B) and the reintroduction sites (D and M), were P$_{tot}$, pH, and N-NH$_4$ (Figures 2 and 3).

More detailed information on the distribution of values for physical-chemical factors of the peatland water was obtained by analyzing data for each of the reintroduction sites M and D in comparison to site B. Depending on the location, similarity, or dissimilarity of values for the habitat parameters was observed in relation to the natural site (B) (Figure 4).

Based on the Kruskal–Wallis test and *p*-values at $p < 0.05$, significant differences in the distribution of values for N-NO$_2$ concentrations in the water were observed for all sites (B, D, and M). In the case of sites M and D, such differences were noted for five parameters, i.e., N$_{tot}$, N-NO$_3$, DOC, EC, and pH, although the distributions of values for N$_{tot}$ concentrations and pH showed significant differences only for the Długie site (D). The distributions of values for N-NH$_4$ and phosphorus fractions did not differ significantly for these sites (Table 1). In the case of site B, statistically, significant differences were noted only for N-NO$_2$, while the remaining physical-chemical factors of the water showed no statistically significant fluctuations (Table 1).

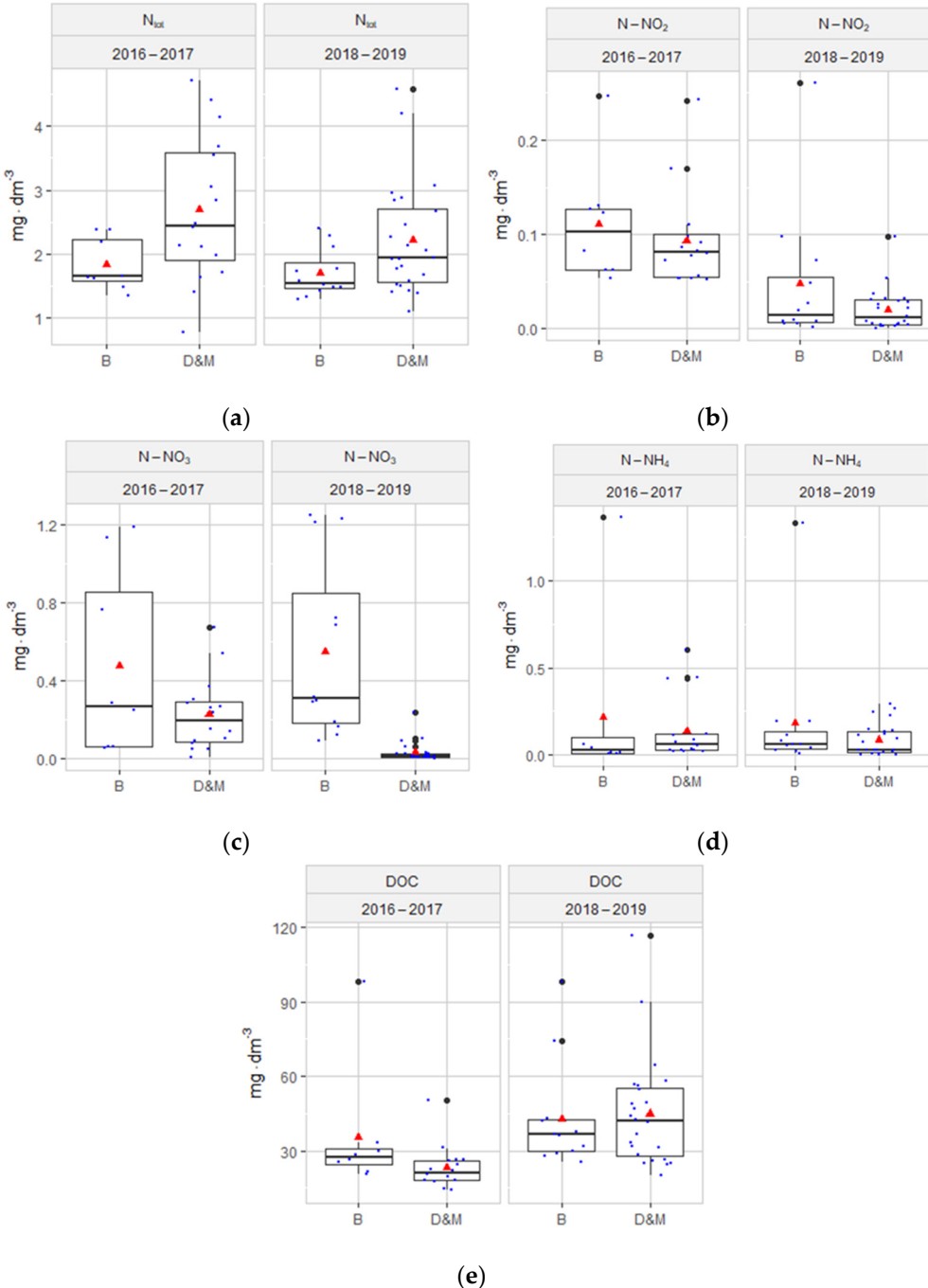

**Figure 2.** Distribution of shallow groundwater values: N-fractions (**a**–**d**) and DOC (**e**) at study sites B, D, and M collectively in 2016–2017 and 2018–2019. Box-and-whisker plots show the distribution of the observations. The box represents the first and third quartiles. The horizontal line across the center of the box represents the median. The mean value of the data is designated by a filled red triangle. The whiskers are drawn to the most extreme observations located no more than 1.5 times the interquartile range away from the box. Any observation not included between the whiskers is considered an outlier and is plotted with a filled circle. When there are no outliers, whiskers indicate the minimum and maximum values. The plot presents observed values of particular parameters, marked with blue dots.

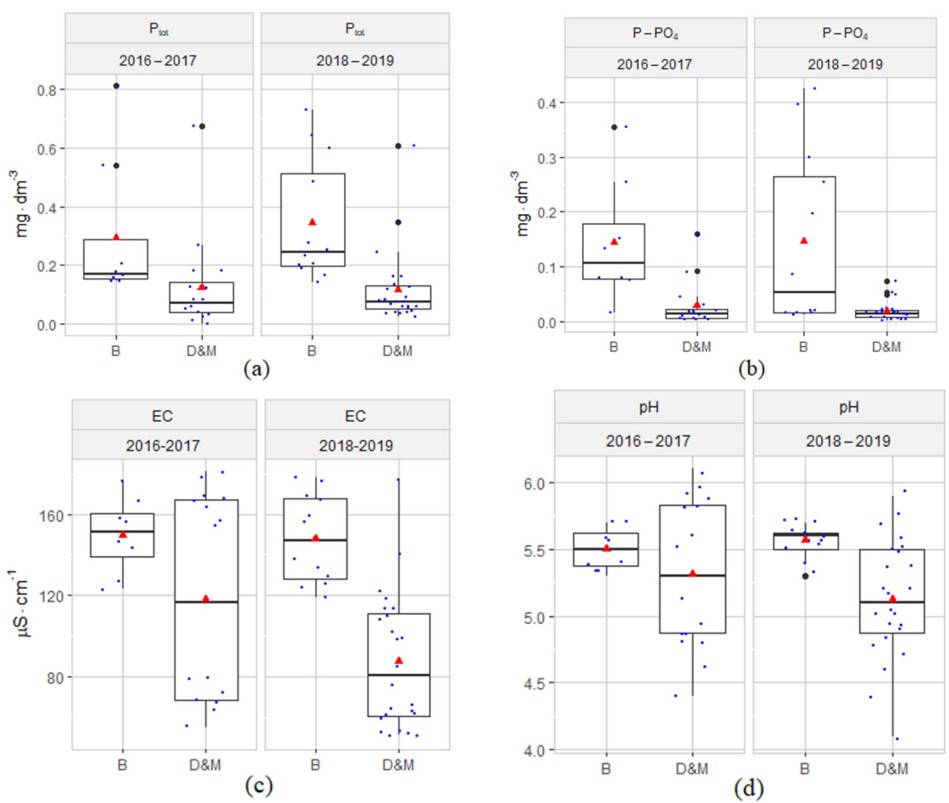

**Figure 3.** Distribution of shallow groundwater values for P-fractions, EC, and pH at study sites B, D, and M collectively in 2016–2017 and 2018–2019. P-fractions (**a**,**b**); EC (**c**); and pH (**d**).

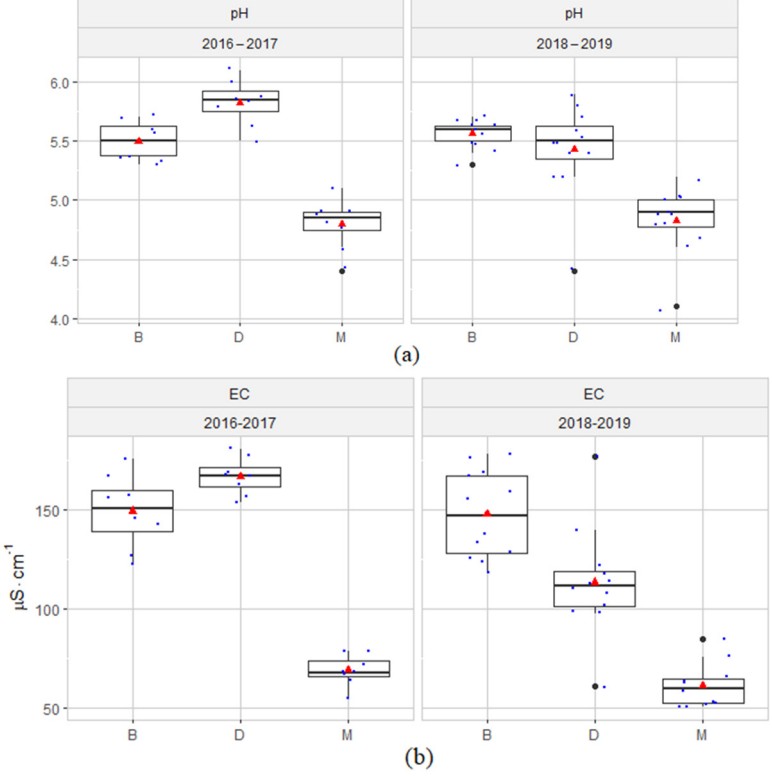

**Figure 4.** *Cont*.

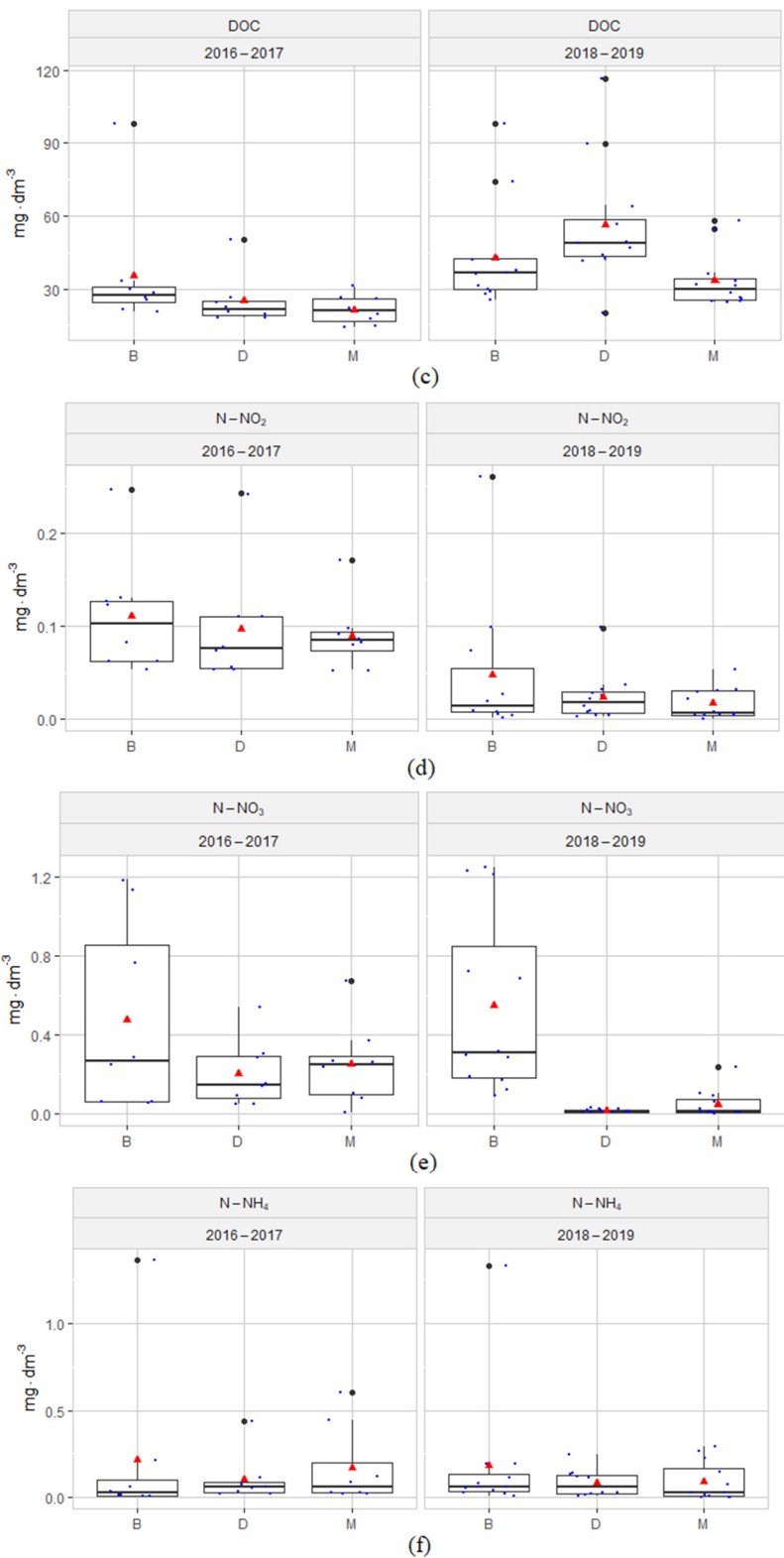

**Figure 4.** *Cont.*

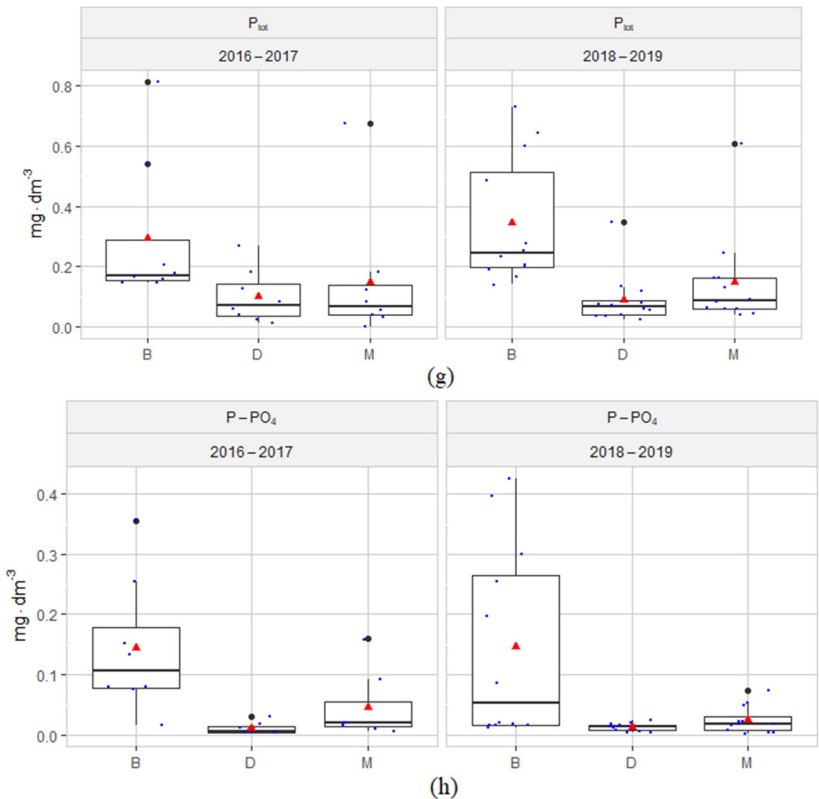

**Figure 4.** Distribution of values of physical-chemical parameters of shallow groundwater at three study sites (B, D, M) in 2016–2017 and 2018–2019. The horizontal line across the center of the box represents the median. The mean value of the data is designated by a filled triangle. Any observation not included between the whiskers is considered an outlier and is represented by a filled circle. (**a**) pH; (**b**) EC; (**c**) DOC; (**d**–**f**) N-fractions; (**g**,**h**) P-fractions.

**Table 1.** Values obtained in the Kruskal–Wallis test and *p*-values for physical-chemical parameters at sites M, D, and B.

| Parameter | Study Site | Kruskal-Wallis Chi-Squared | *p*-Value |
|---|---|---|---|
| $N_{tot}$ | B | 0.788 | 0.3746 |
| | D | 4.339 | 0.0372 |
| | M | 0.054 | 0.8170 |
| $N\text{-}NO_3$ | B | 0.932 | 0.3345 |
| | D | 13.735 | 0.0002 |
| | M | 7.319 | 0.0068 |
| $N\text{-}NH_4$ | B | 0.860 | 0.3536 |
| | D | 0.013 | 0.9078 |
| | M | 1.526 | 0.2167 |
| $N\text{-}NO_2$ | B | 6.487 | 0.0109 |
| | D | 11.031 | 0.0009 |
| | M | 13.208 | 0.0003 |

**Table 1.** *Cont.*

| Parameter | Study Site | Kruskal-Wallis Chi-Squared | *p*-Value |
|---|---|---|---|
| $P_{tot}$ | B | 1.527 | 0.2165 |
| | D | 0.0536 | 0.817 |
| | M | 0.788 | 0.3746 |
| $P\text{-}PO_4$ | B | 0.150 | 0.6984 |
| | D | 0.866 | 0.3522 |
| | M | 0.658 | 0.4174 |
| pH | B | 0.636 | 0.4252 |
| | D | 7.409 | 0.0065 |
| | M | 0.303 | 0.5820 |
| EC | B | 0.002 | 0.9692 |
| | D | 10.500 | 0.0012 |
| | M | 4.519 | 0.0335 |
| DOC | B | 3.434 | 0.0639 |
| | D | 8.149 | 0.0043 |
| | M | 6.881 | 0.0087 |

Based on the correlation matrix, strong or moderate correlations, only positive, were shown between values for physical-chemical parameters of the water, e.g., $P_{tot}$ with $P\text{-}PO_4$, $P_{tot}$ with $N\text{-}NO_3$, and electrical conductivity with pH (Figure 5).

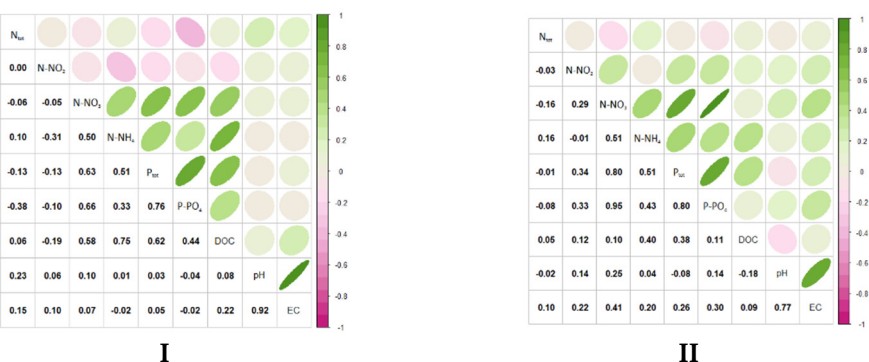

**Figure 5.** Correlations between physical-chemical factors of peatland water as pooled data for all sites in 2016–2017 ((**I**)–first study period) and 2018–2019 ((**II**)–second study period).

The statistical analyses were supplemented with indirect ordination methods (PCA) aimed at detecting the structure and general patterns between the analyzed physical-chemical water parameters at individual sites.

A fairly strong positive correlation was noted between EC and pH and between DOC and $P_{tot}$, as well as a strong negative correlation between $N_{tot}$ and $P_{tot}$. The correlation between $N_{tot}$ and EC and the correlation between pH and $P_{tot}$ were close to zero (Figure 6).

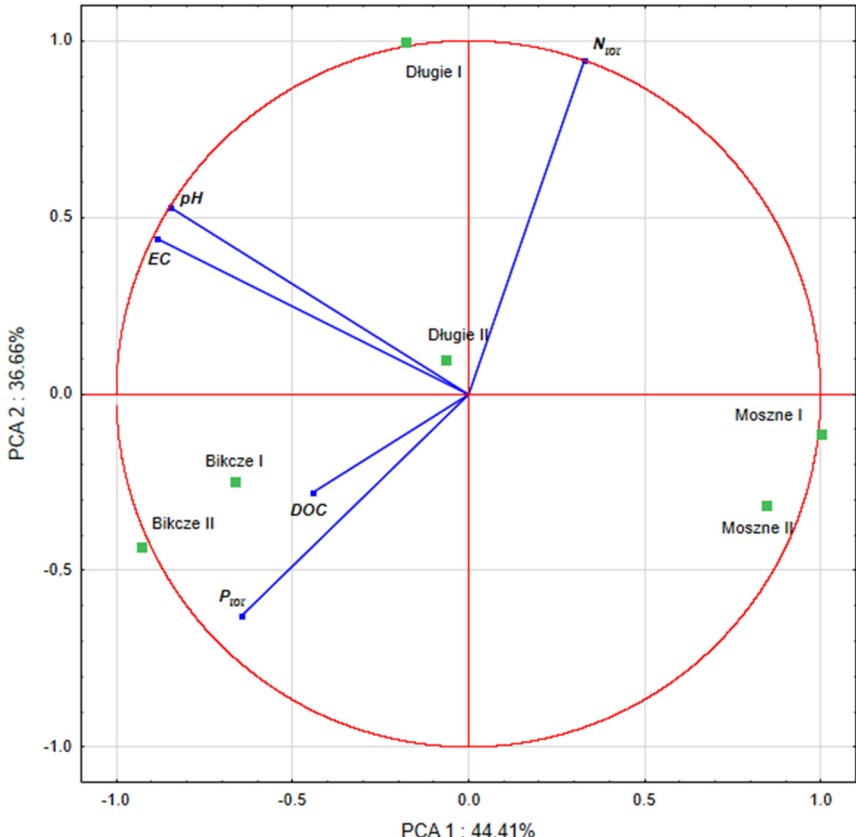

**Figure 6.** Principal component analysis (PCA) of physical-chemical parameters of groundwater for sites Bikcze, Moszne, and Długie in both study periods (I and II).

Analysis of the physical-chemical factors of the water indicates differing conditions in the habitats of sites M and B (in both study periods), especially in the case of pH, electrical conductivity, organic carbon content, and total phosphorus content. However, the conditions at these sites did not change significantly between the two study periods, in contrast to site D, where marked changes in the parameters were noted (Figure 6).

### 3.2. Biocoenotic Conditions of the Habitats

The next step of the research was confirmation of the biocoenotic stability of the habitats at the sites based on changes in the species composition of the phytocoenoses and microbiological characterization of the soil.

At the reintroduction sites (D and M) and the comparison site (B), a total of 49 plant species belonging to 24 botanical families were identified in the two study periods (Table 2).

**Table 2.** Species composition of the research sites (M, D, and B) in the years 2016–2017 (I) and 2018–2019 (II).

| CLASS | MAGNOLIOPSIDA | I | | | II | | |
|---|---|---|---|---|---|---|---|
| | | **M** | **D** | **B** | **M** | **D** | **B** |
| **Botanical Family** | **Species** | | | | | | |
| Apiaceae | *Peucedanum palustre* | + | + | | | | + |
| | *Alnus glutinosa* | + | | | | | |
| | *Alnus incana* | | + | | | + | |
| Betulaceae | *Betula humilis* | + | + | + | | | |
| | *Betula pendula* | | | | + | | + |
| | *Betula pubescens* | + | + | + | + | + | + |

**Table 2.** *Cont.*

| Botanical family | Species | I M | I D | I B | II M | II D | II B |
|---|---|:--:|:--:|:--:|:--:|:--:|:--:|
| Caryophyllaceae | *Stellaria palustris* | | | + | | | |
| Droseraceae | *Drosera rotundifolia* | + | + | | + | + | |
| Ericaceae | *Andromeda polifolia* | + | + | | + | | + |
| Ericaceae | *Oxycoccus palustris* | + | + | + | + | + | + |
| Lythraceae | *Lythrum salicaria* | + | | | | | |
| Menyanthaceae | *Menyanthes trifoliata* | + | + | + | + | + | + |
| Parnassicaceae | *Parnassia palustris* | | + | | | | |
| Primulaceae | *Lysimachia thyrsiflora* | | | + | | | |
| Primulaceae | *Lysimachia vulgaris* | | + | + | | | + |
| Ranunculaceae | *Ranunculus lingua* | | | + | | | |
| Rhamnaceae | *Frangula alnus* | | + | | | | + |
| Rubiaceae | *Galium palustre* | + | + | | | | |
| Rosaceae | *Comarum palustre* | + | + | + | | | + |
| Rosaceae | *Potentilla erecta* | | + | | | | |
| Salicaceae | *Salix cinerea* | + | + | + | + | + | + |
| Salicaceae | *Salix lapponum* | + | + | + | + | + | + |
| Salicaceae | *Salix mytrylloides* | | + | | | + | + |
| Salicaceae | *Salix rosmarinifolia* | + | | | | | |
| Scheuchzeriaceae | *Scheuchzeria palustris* | | | | + | + | |
| **CLASS** | **LILIOPSIDA** | **I** | | | **II** | | |
| **Botanical family** | **Species** | **M** | **D** | **B** | **M** | **D** | **B** |
| Araceae | *Calla palustris* | | | + | | | |
| Cyperaceae | *Carex acutiformis* | | | + | | | |
| Cyperaceae | *Carex appropinquata* | | | | + | + | |
| Cyperaceae | *Carex curta* | | | | | | + |
| Cyperaceae | *Carex echinata* | + | + | | | | |
| Cyperaceae | *Carex elata* | | + | | | | |
| Cyperaceae | *Carex lasiocarpa* | + | + | + | | | + |
| Cyperaceae | *Carex limosa* | + | + | | | | |
| Cyperaceae | *Carex nigra* | + | | | | | + |
| Cyperaceae | *Carex panicea* | + | + | | | | + |
| Cyperaceae | *Carex rostrata* | + | | + | + | + | + |
| Cyperaceae | *Eriophorum angustifolium* | + | + | | | | + |
| Cyperaceae | *Eriophorum vaginatum* | + | | | | | |
| Cyperaceae | *Rhynchospora alba* | + | | | | | |
| Orchidaceae | *Dactylorhiza incarnata* | | + | | + | + | |
| Poaceae | *Calamagrostis canescens* | | | | | | + |
| Poaceae | *Molinia caerulea* | + | + | | | | |
| Poaceae | *Phragmites australis* | + | | | | | |
| Typhaceae | *Typha latifolia* | | | + | | | |

**Table 2.** *Cont.*

| CLASS | EQUISETOPSIDA | I | | | II | | |
|---|---|---|---|---|---|---|---|
| | | M | D | B | M | D | B |
| **Botanical family** | **Species** | | | | | | |
| Equisetaceae | *Equisetum fluviatile* | | | | | + | |
| | *Equisetum palustre* | + | + | + | | | + |
| **CLASS** | **POLIPODIOPSIDA** | I | | | II | | |
| | | M | D | B | M | D | B |
| **Botanical family** | **Species** | | | | | | |
| Thelypteriodaceae | *Thelypteris palustris* | + | + | + | + | + | + |
| **CLASS** | **CONIFEROPSIDA** | I | | | II | | |
| | | M | D | B | M | D | B |
| **Botanical family** | **Species** | | | | | | |
| Pinaceae | *Pinus sylvestris* | + | + | | + | + | |
| **CLASS** | **SPHAGNOPSIDA** | I | | | II | | |
| | | M | D | B | M | D | B |
| **Botanical family** | **Species** | | | | | | |
| Sphagnaceae | *Sphagnum* sp. | + | + | + | + | + | + |
| Total | 49 | 29 | 30 | 19 | 14 | 15 | 22 |
| | | | 44 | | | | 29 |

In 2016–2017 the highest number of plant species was identified at site D—30 species, and the fewest next to Lake Bikcze (B)—19 species. In the second study period (2018–2019) it was site B that had the highest species richness (Table 2).

Jaccard's species similarity index was similar in both study periods (I and II). It was highest for the comparison between the phytocoenoses of sites D and M (first study period: $S_{M\text{-}D} = 0.513$; second study period: $S_{M\text{-}D} = 0.75$). Lower values ($S < 0.5$) were noted when comparing the phytocoenoses of the two reintroduction sites of *S. lapponum* with the natural site (B; Table 3).

**Table 3.** Values for Jaccard's species similarity index for the research sites in 2016–2017 (I) and 2018–2019 (II). Extreme values are indicated by colors: yellow, min; orange, max.

| I | | | II | | |
|---|---|---|---|---|---|
| **M–D** | **M–B** | **D–B** | **M–D** | **M–B** | **D–B** |
| 0.513 | 0.375 | 0.333 | 0.75 | 0.36 | 0.321 |
| | **TOTAL** | | | | |
| **M–M** | **B–B** | **D–D** | | | |
| 0.37 | 0.5 | 0.363 | | | |

The most stable species composition during the study was recorded for site B ($S_B = 0.5$; Table 3), as confirmed by analysis of the dendrogram based on Jaccard's species similarity index combined with a heatmap assigning the frequency of co-occurrence of identified plant species to phytocoenoses of sites in successive years (Figure 7).

Based on the analyses of the species similarity of the flora, two clusters were distinguished. The first comprised the vegetation of sites M and D in the years 2018–2019, while the second contained the flora of site B in both study periods (I and II) and of sites M and D in the years 2016–2017 (Figure 7).

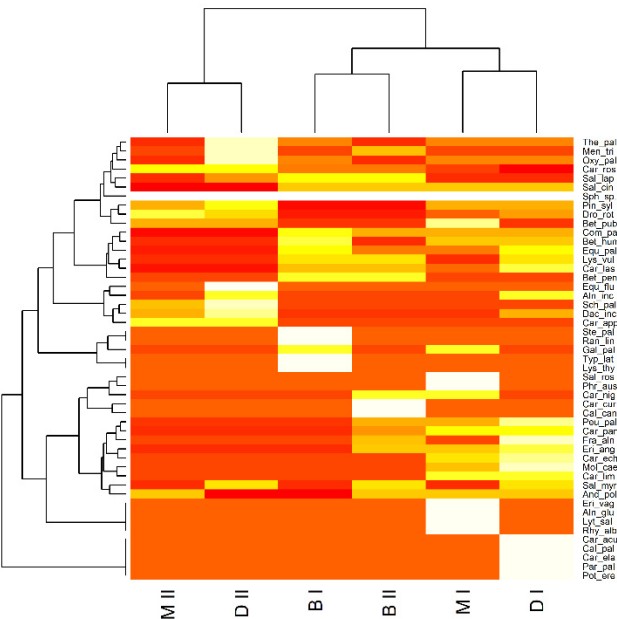

**Figure 7.** Dendrogram and heatmap of the hierarchical cluster analysis of 49 species (listed in Table 2) and three research sites (M, B, D) in 2016–2017 (I) and 2018–2019 (II) based on Jaccard's index and the average linkage method (white and yellow indicate the presence of a given species in the phytocoenoses; red indicates its absence).

More clusters can be observed in the dendrogram describing the tendency towards co-occurrence of individual plant species. The most visible are eight groupings on the lower level of the dendrogram (Figure 7). An example is the grouping of the species *Carex acutiformis*, *Calla palustris*, *Carex elata*, *Parnassia palustris*, and *Potentilla erecta*, present in its entirety at site D in 2016–2017 but at neither of the other sites (Table 2, Figure 7). Another example is the group consisting of the *Eriophorum vaginatum*, *Alnus glutinosa, Lythrum salicaria,* and *Rhynchospora alba*, which was present in full only at site M in 2016–2017 (Table 2, Figure 7).

Plant species in individual groups belonged to a variety of syntaxonomic units, sometimes determining different habitat types (raised bogs, transitional bogs, and fens).

The microbiological analyses showed that in the case of bacteria, the number of colony-forming units (CFU) per g DW was lowest in the soil sampled from the rhizosphere of the reintroduced plants on the peatland near Lake Długie (D). The highest bacterial count was noted for the rhizosphere soil from the site near Lake Moszne (M) (Table 4).

**Table 4.** Numbers of bacteria and fungi per g of dry weight of soil within and outside the rhizosphere at sites M and D in 2018–2019.

| Parameter/Soil | Research Site | | | |
| --- | --- | --- | --- | --- |
| | M | | D | |
| | Rhizosphere | Non-RhizoSphere | Rhizosphere | Non-RhizoSphere |
| Dry weight [g] | 0.7 | 0.11 | 0.53 | 0.44 |
| Bacterial count [CFU $\times$ g$^{-1}$ g DW] | $51.43 \times 10^{-6}$ | $30.08 \times 10^{-6}$ | $2.13 \times 10^{-6}$ | $8.54 \times 10^{-6}$ |
| Fungal [CFU $\times$ g$^{-1}$ g DW] | $225.71 \times 10^{-3}$ | $186.22 \times 10^{-3}$ | $254.1 \times 10^{-3}$ | $226.33 \times 10^{-3}$ |

The number of fungi in 1 g of dry weight of soil (CFU/g DW) varied between sites. The most fungal colonies were isolated from the rhizosphere soil of downy willow individuals growing on the peatland near Lake Długie (site D), and the fewest from the soil outside the rhizosphere of the plants at site M (Table 4).

The species composition of fungi isolated from the rhizosphere and non-rhizosphere soil at the two sites was similar. The total number of identified fungal species for both categories of samples and the two sites (D and M) was 8 species from 4 families.

The most frequently isolated fungi belonged to the genus *Trichoderma* of the family Hypocreaceae. These were the species *T. hamatum*, *T. polysporum*, *T. aureoviride*, and *T. konigii.* Species of the family Aspergillaceae were also frequently isolated, including *Aspergillus niger* and *Talaromyces funiculosus* as well as less abundant species of the genus Aspergillus. The fungal populations also included single isolates of fungi of the families Pleosporaceae and Sarocladiaceae: *Alternaria alternara* and *Sarocladium strictum*, respectively (Figure 8).

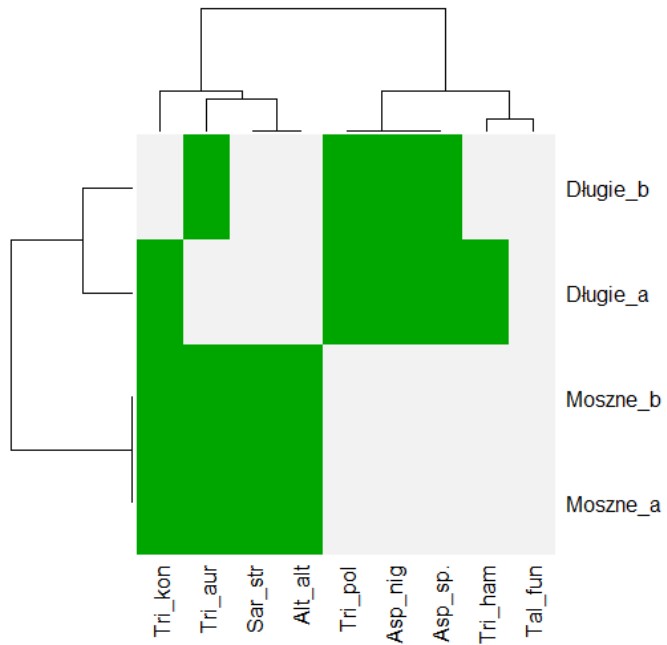

**Figure 8.** Dendrogram and heatmap of the hierarchical cluster analysis of 9 species of soil fungi and two research sites (M, D) for soil from the rhizosphere (a) and outside the rhizosphere (b) based on the average linkage method (the presence of a given species is indicated by the color grey and its absence by the color green). List of species: *Alternaria alternata* (Fr.) Keissl., *Aspergillus niger* Tiegh., *Aspergillus* spp., *Sarocladium strictum* (W. Gams) Summerb., *Talaromyces funiculosus* (Thom) Samson, N. Yilmaz, Frisvad & Seifert, *Trichoderma aureoviride* Rifai, *T. hamatum* (Bonord.) Bainier, *T. kongii* Oudem., *T. polysporum* (Link.) Rifai.

Analysis of the species composition of fungi isolated from the rhizosphere and non-rhizosphere soil of plants growing at site M showed that the same species were obtained from both soil zones. At site D, however, the species *T. aureoviride* was present in the rhizosphere but not outside the rhizosphere of downy willow, where there were also two species recorded that were not present in the rhizosphere: *T. hamatum* and *T. konigii*. A comparison of the two sites revealed differences in the species composition of the two zones. In the case of the rhizosphere, the only species present at both sites was *Talaromyces funiculosus*, while in the non-rhizosphere soil the species *T. hamatum* was common at both sites (Figure 8).

These observations are confirmed by a more detailed analysis of the combined dendrogram and heatmap (Figure 8). At the level of species similarity, we can distinguish two groups: 1—in the rhizosphere (a) and outside the rhizosphere (b) at site M, with identical species composition of soil fungi; 2—in the same zones at site D (a and b), more varied in this respect.

## 4. Discussion

Ecological relationships, which are crucial for the functioning of natural ecosystems, are directly dependent on abiotic conditions, including the state and qualitative parameters of groundwater. Multi-faceted monitoring conducted to determine the effect of habitat quality on the structure and functioning of populations of rare and endangered relict plant species are particularly important for the effectiveness of active measures for their conservation [3,4,13,20,28,35–38].

The preservation or restoration of endangered populations of wetland plant species requires a thorough knowledge of their biology. The effect of various physical-chemical factors of the groundwater on the condition, growth, and development of plants is also of great importance in this regard [38–42]. In the peatlands of eastern Poland, this problem particularly affects Pleistocene boreal relicts such as *Betula humilis*, *Salix myrtylloides*, and *Salix lapponum* [4,17,28,43].

The search for a suitable location for reintroduction should be based on knowledge of the species' habitat preferences, but also knowledge of potential future transformations of the habitats [21]. To predict these, it is essential to monitor the abiotic and biotic conditions of habitats and analyze changes in the physical-chemical factors of surface water quality in relation to qualitative and quantitative analyses of flora [23–25].

The study sites where *Salix lapponum* was reintroduced are located on land covered by the protection of natural areas, in the form of a national park. They contained peatland habitats on organogenic soil, whose hydrochemical characteristics reflected their typology and internal metabolism, and also the effect of external, anthropogenic factors. This effect is directly and indirectly associated with non-organized tourism, extensive agriculture, and especially drainage lowering the groundwater table throughout the region. Changes in hydrological conditions in the study area are associated with the operation of the drainage system of the Wieprz–Krzna Canal. The local effect of mining in the Lublin Coal Basin also plays an important role [3,4,44].

### 4.1. Hydrochemical Conditions of the Habitats

The chemical composition and thus the quality of peatland water depends on multiple factors, of which the most important are the geological and topographic features of the region (i.e., surface runoff and landscape retention) [45] and climate conditions (temperature, moisture, and air pressure). Other significant factors are the chemical composition and distribution of precipitation, local and regional water management, the type and intensity of use of the area, and the type of vegetation [46,47].

The geological structure of the Polesie Podlasie region of Eastern Poland is the opposite of the monotonous terrain. The chalk bedrock resting on Palaeozoic formations in the form of limestones, marls, and Jurassic and Cretaceous rocks is the basis for Tertiary and Quaternary sediments. They are composed mainly of clays and gravels with sands of glaciation moraines, and the formation of water, river, and lake accumulation, including silt and peat. Hipsometrically, this region represents a vast morphological depression consisting of accumulation and denudation plains with sparse chalk remnants in the form of low, gentle chalk hills. These rocks appear directly on the surface or are covered with a cover of glacial formations, they are made of not very resistant marls and are easy to weather [28]. The great diversity of the soils of the studied area is caused by the quality of Quaternary sediments. And so, in the areas dominated by loess and loess formations, there are soils of the brown-earth type—mainly loaches. The areas formed by sandy and clay sediments were mainly formed by podzolic soils of various types. On the other hand, in peat bogs, one can find soils belonging to the classes of marshy and post-bog soils of various types. Cretaceous outcrops in the described area produced Cretaceous rendzinas [28].

The climatic conditions of the region belong to the group of the Great Valleys climate, although they are distinguished by a large number of continental features, such as long summer—105 days and winter—110 days, short spring and autumn—often less than 50 days. Throughout the year, polar-sea air masses predominate. The average annual

amount of precipitation only slightly exceeds the strong evaporation (humid climate), which makes the region poorer in terms of the possibility of renewing water resources. Precipitation is characterized by a very high spatial and temporal variability. In dry years, the amount of precipitation may fall below 400 mm, and in wet years it may exceed 850 mm. Most precipitation falls in summer (approx. 40%, torrential precipitation), winter is a period of low precipitation, therefore the snow cover reaches an average height of only 12 cm [48].

The area of Eastern Poland includes the Polesie Podlaskie Region with the Łęczyńsko-Włodawskie Lakeland—an area unique on a European scale in terms of nature and landscape. The most valuable elements here are lake and peat bog complexes (raised bogs, transitional bogs, and low bogs), which at the same time belong to the most sensitive ecosystems susceptible to changes. Forest areas occur only locally and form the following complexes: Włodawskie, Parczewskie, and Sobiborskie Forests. Due to the poor habitats, pine forests appear here in fresh and moist habitats, less often—pine-oak forests and mixed forests, which are much more ecologically valuable. More fertile habitats are oak-hornbeam forests, and wetlands are covered by alder forests and forest alder [28].

The fundamental changes in the chemical composition of peatland water in Western Polesie are of anthropogenic origin and are mainly the result of changes in hydrologic conditions (drainage). Drainage of peat deposits results in the mineralization of the organic matter contained in surface formations. Previously deposited organic matter gradually disappears, releasing large amounts of nutrients (mainly nitrogen and sulfur). Acidification of peat soil is associated with the loss of basic cations due to phytoretention and leaching from the soil profile [49,50]. The reduction in pH causes a gradual release of phosphorus due to increased solubility of its compounds (e.g., variscite). This leads to the activation of aluminum, which is toxic for plants [51,52]. The increased content of highly chemically active phosphorus in peat soils poses a serious threat to its dispersion into the water. This causes changes in chemical characteristics and affects the trophic status of many ecosystems [53]. Groundwater contamination accelerates the eutrophication of surface waters, additionally fed by surface and drainage runoff [47]. Eutrophication of peatland habitats also frequently results from changes in the water regime associated with hydrotechnical modifications and influx of nutrients of municipal origin, from agriculture or other branches of the economy, or atmospheric deposition [52,54]. An increase in trophy and thus the productivity of biocoenoses induces ecological succession in natural ecosystems [3].

The content of nutrients, i.e., nitrogen and phosphorus, which determines the quality of shallow groundwater, can be a potential indicator of non-point source pollution, including agricultural pollution [55–57].

In 2016–2019, at the sites of reintroduction of *S. lapponum*, changes were observed in the concentration of the nitrogen and phosphorus fractions in the groundwater. Their average values were beyond the ranges of the typical distribution of observations specified by the median. They did not, however, exceed the values characteristic of mesotrophic habitats. A difference was also noted between the two study periods in the values of EC, $N_{tot}$, $N$-$NO_3$, and DOC, which may indicate the appearance of eutrophic water in the study area. However, the stability of parameters such as pH, $N_{tot}$, and $N$-$NH_4$ suggests that no significant changes in hydrographic conditions were taking place [49,50]. This state may also be the result of the temporary mobilization of biogenic compounds within the peatland resulting from mechanisms of its metabolism [58].

Irrespective of the genesis of the changes described above, the values of the physical-chemical parameters of the peatland water were not outside the range of ecological valence of *S. lapponum* [11,19]. This confirms the hypothesis that habitats suitable for glacial relict plants in terms of abiotic conditions may also be located beyond the limits of the species' range of occurrence [20,59]. Analysis of the physical-chemical factors of the water at the sites of reintroduction of *S. lapponum* (D and M) in comparison to the site of its natural occurrence (B) revealed marked differences in the pH and electrical conductivity of the groundwater in 2016–2017 and additionally in the content of organic carbon, total phos-

phorus and ammonium ions in 2018–2019. Evaluation of the stability of new populations will answer the question of whether the physical-chemical factors of the environment will influence the success of the full life cycle of the species.

The pH of the groundwater at the sites of *S. lapponum* was slightly acidic and characteristic of the transitional bogs and fens in the Łęczna-Włodawa Lakeland. The measurement results did not indicate that secondary anthropogenic eutrophication of these ecosystems was taking place [50]. Similarly, the values for electrical conductivity, an indirect measure of water mineralization and pollution, were not indicative of a significant increase in processes associated with human impact.

The fundamental factor accelerating the eutrophication of groundwater in peatlands is the quantity and quality of dissolved organic matter (DOC). This determines the availability of easily assimilated forms of nitrogen and phosphorus bound to humic substances [58]. However, the values of this parameter observed in 2016–2019 did not indicate the possibility of secondary activation of biogenic compounds for plants at the study sites. The absence of significant human impact on the habitats is confirmed by the typical positive correlations between nutrient fractions—mainly $P_{tot}$ with $P-PO_4$, $N-NO_3$ with DOC, and electrical conductivity with pH. These may indicate that the mechanisms of the internal metabolism of the peatland were undisturbed.

The values of the physical-chemical parameters of the shallow peatland water at the sites of reintroduction of *S. lapponum* (D and M), despite temporary fluctuations due to mechanisms of the internal metabolism of the bog, should not be a significant factor limiting the stability of the new populations of this species. Nevertheless, the natural site of the downy willow population near Lake Bikcze was much more hydrochemical stable; statistically significant differences were noted only for $N-NO_2$.

### 4.2. Biocoenotic Conditions of the Habitats

The chemical composition of shallow peatland water, directly and indirectly, affects the structure and functioning of biocoenoses. Primary or secondary eutrophication of water results in more homogeneous biocoenoses due to a rapid increase in the biomass of highly competitive plants and thus a loss of biological diversity [52]. The intensification of secondary succession processes may in the long-term lead to the transformation of peatland ecosystems with low vegetation into forest communities [3,60–62]. These changes lead to the loss of local sites of plant species characteristic of particular types of peatlands. The problem significantly affects rare and protected Pleistocene boreal relicts [3,13,14,17,19,43,59]. An example is the species *S. lapponum*, whose population size in Poland, relative to data from the 1950s, has decreased by 80% [11,27].

The species composition and life forms of plants and the mosaic-like nature of peatland phytocoenoses secondarily determine the potential of the biogeochemical barrier of plant communities. They also affect the retention of nutrients and other contaminants [38]. The state of biocoenoses at the sites of reintroduction of *S. lapponum* is therefore of fundamental importance for the stability of new populations of this species.

In 2016–2017 and 2018–2019, a total of 49 plant species belonging to 24 botanical families were identified at the study sites. All the species are characteristic of peatland habitats undisturbed by human impact [19], which suggests that the flora of these ecosystems is to a large extent natural. Slightly different conclusions can be drawn regarding the biocoenotic stability of these habitats. In 2016–2017 species richness was lowest at the site of the natural occurrence of *S. lapponum* near Lake Bikcze (B—19 species) and highest at the site on the peatland near Lake Długie (D—30 species). In the second study period (2018–2019), fewer total plant species were identified at the site near Lake Moszne (M—14 species) and the most at the site near Lake Bikcze (B—22 species). Changes in species composition can be the first signal indicating hydrological disturbances caused by human activity [63]. This was also confirmed by the analysis of Jaccard's species similarity index. Despite the relatively high species similarity between the reintroduction sites (M and D) for both study periods ($S_{M-D}$ = 0.53 and 0.75), considerable changes were observed over time at each of the sites.

In syntaxonomic terms, most of the plants at the study sites are represented by the alliance *Magnocarition* and the classes *Scheuzerio-Caricetea* (mainly of the order *Caricetalia nigrae*), *Alnetea glutinosae,* and *Oxycoco-Sphagnetea*. The alliance *Magnocarition* belongs to the class *Phragmitetea* and encompasses plant communities of hygrophytes forming reed beds on the shores of running and standing water bodies and to a lesser degree in the littoral, occupying the zone separating *Phragmition* from peatland vegetation, wet meadows, and forests. They mainly consist of sedges and are present in fens and transitional bogs, with deposits of peat derived from sedges or reeds and sedges. The class *Scheuzerio-Caricetea* consists of communities of wet meadows with low sedges and bryophytes. The class *Alnetea glutinosae* is formed of alder carrs and wet willow thickets, which are floristically and ecologically similar to them. These associations develop on wet peat soils or peat and mineral soils under the influence of a high groundwater level with obstructed outflow, which at certain times of year rises to the surface and floods small depressions. The class *Oxycoco-Sphagnetea* consists of communities of dwarf shrubs and peat moss of wet heathlands and raised bogs in acidic oligotrophic and dystrophic habitats fed mainly by rainwater [64].

The combinations of species characteristic of fens and raised bogs are indicative of habitats of an intermediate character [19]. This is confirmed by the characteristic communities aggregating indicator and companion species characteristic of raised bogs, fens, and transitional bogs co-existing at the locations of the reintroduction sites (M and D) and the natural site (B). The presence in the phytocoenoses of various species of trees and shrubs of the family Betulaceae (5 species), for example, may indicate natural processes of ecological succession associated with the development of the phytocoenosis. In conclusion, the species composition of the flora accompanying *S. lapponum* reflects the characteristic features of the habitat, which in this case do not indicate significant anthropogenic disturbances.

The biological activity of the soil is an important factor in determining the success of the reintroduction of endangered plant species. It influences soil fertility and plants' access to nutrients [63]. The microbiological activity of the soil, particularly rhizosphere microbes, is a factor determining the optimal growth of plants and their resistance to stress, diseases, pests, and weeds. Rhizosphere microbes can be neutral or can have positive or negative effects on plants [65–67]. The species diversity of soil microbes, which depends on multiple physical and chemical properties of the soil, climate factors, and plant root secretions, is indicative of the richness and fertility of the soil [68,69].

The microbiological analyses of the soil conducted at the sites of reintroduction of *S. lapponum* showed varied numbers of colony-forming units of bacteria and fungi within and outside the rhizosphere of the species. According to literature reports, the number of bacteria in 1 g of soil can be as high as $5 \times 10^7$ cells [70]. The results of our analyses showed that the rhizosphere of *S. lapponum* is poor in bacterial cells, most likely due to the acidic reaction of the soil. According to Galus-Barchan and Paśmionka [69], bacteria prefer an alkaline environment—ideally soil with a pH from 6.5 to 7.5. This explains the low abundance of bacteria, as the conditions in the soil were not highly favorable to them. However, the lowest numbers of bacteria within and outside the rhizosphere were noted at site D, where the pH was slightly higher than at the other (M) site of the reintroduction of *S. lapponum*. Nevertheless, it was much higher than pH = 6. Therefore, there must be other factors determining the lower number of bacteria at this site. The species composition of fungi, in both types of samples, was similar for the two sites of reintroduction of downy willow, although the highest number of colony-forming units per g DW of soil was noted at site D.

Analysis of the species composition of fungi showed no pathogenic species, e.g., of the genus *Fusarium*, which pose a threat to both the above-ground and below-ground parts of many plant species, including plants of the genus *Salix* [71,72]. A positive finding was the occurrence of saprotrophic species, which contribute to the decomposition of organic matter, and of hyperparasitic fungi of the genus *Trichoderma*, exploited in biological plant protection. Fungal species of this genus produce amylase, cellulase, phosphatase, xylanase, lipase, and protease, thereby contributing to the decomposition of organic matter [72,73].

The literature lacks information on soil microbes forming the population of bacteria and fungi of the rhizosphere of downy willow, but the analyzed species variation of the mycoflora is most likely indicative of normally functioning processes of decomposition of organic substances in the soil. This suggests that nutrients are easily accessible to plants, and thus that the reintroduction of the plant to the natural environment will be successful.

**5. Conclusions**

The results of laboratory and field studies allowed for the conclusion that:

1.　The hydrochemical characterization of the peatland groundwater at the sites of reintroduction of *S. lapponum* and changes in the parameters over time does not indicate an intensifying influx of nutrients or other hydrological disturbances caused by human activity.
2.　Although in 2016–2019 the values of the physical-chemical factors of the groundwater showed variability associated with the internal metabolism of the peatlands, they were within the range of the habitat preferences of the species *S. lapponum*.
3.　The minor changes in the abiotic factors of the environment observed during the study should not have a limiting effect on the *S. lapponum* population.
4.　The qualitative state of the phytocoenoses and the changes noted over the 4 years of the study at the sites where *S. lapponum* individuals were planted do not indicate any disturbances caused by human activity. It should be borne in mind, however, that hydrological disturbances in the habitats could accelerate processes of ecological succession associated with changes in the species composition and quantitative structure of the flora.
5.　Biological activity and the absence of pathogenic fungi in the soil of the sites where plants were reintroduced indicate normal functioning of soil processes, and if this biological activity is not disturbed in the near future, it should be favorable to active conservation of *S. lapponum*.
6.　The new *S. lapponum* populations can be expected to survive and develop at the study sites provided that the current hydrochemical stability and thus the biocoenotic stability of the habitats are maintained.

The correct selection of habitats for the reintroduction of endangered plant species and the problem of maintaining the sustainability of their populations require constant monitoring of environmental factors (biological, physical, and chemical), which determines the success of conservation measures for these species.

**Author Contributions:** Conceptualization, A.S.; data curation, A.S.; formal analysis, B.B.-A., E.Z. and U.B.-M.; funding acquisition, M.P.; investigation, A.S., M.P., B.B.-A., E.Z. and M.A.; methodology, A.S. and M.P.; project administration, M.P.; resources, A.S. and M.P.; software, U.B.-M.; supervision, A.S.; validation, A.S., M.P. and U.B.-M.; visualization, U.B.-M.; writing—original draft, A.S.; writing—review & editing, M.P. All authors have read and agreed to the published version of the manuscript.

**Funding:** This research was funded in part by the European Union through the Infrastructure and Environment Operational Programme, project title: Ochrona czynna szczególnie zagrożonych gatunków roślin reliktowych z rodziny Salicaceae w siedliskach torfowiskowych (Active conservation of endangered relict plant species of the Salicaceae family in peat bog habitats), no. POIS.02.04.00-00-0008/17.

**Institutional Review Board Statement:** Not applicable.

**Informed Consent Statement:** Informed consent was obtained from all subjects involved in the study.

**Data Availability Statement:** Not applicable.

**Acknowledgments:** We would like to thank Agnieszka Szczurowska and Michał Niedźwiecki for technical and field studies support.

**Conflicts of Interest:** The authors declare no conflict of interest.

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
