# Peer review of "The Importance of Groundwater Quality and Other Habitat Parameters for Effective Active Protection of an Endangered Plant Species in Eastern Poland"

_water, doi:10.3390/w14081270_

Round 1

Reviewer 1 Report

General comments

The paper is generally well written. You need to insert more details on the aquifer-type, study site and sampling methodology. Please, follow my specific comments to fix the issues. Note that, all the comments need to be addressed before publication

Specific comments

Abstract

Line 18. “Hydrographic conditions”, the expression is difficult to understand to hydrologists that dominate the audience of this journal. Possible a change? Aside from this point, the abstract is very well written

Introduction

Line 51. “Water quality”. I suggest that you specify that you refer to both natural surface water (or soil water) and groundwater

Lines 51-52. Insert most recent hydrological research on surface water-groundwater interaction that deal with crops/plants and incorporate physio-chemical properties (T, EC and PH), nitrate, DOC and major ions/cations as in your research. See reference below

Medici, G., P. Baják, L. J. West, P. J. Chapman, and S. A. Banwart. 2021. DOC and nitrate fluxes from farmland; impact on a dolostone aquifer KCZ. Journal of Hydrology 595, 125658.

Lines 120-171. Integrate other publications on livestock and groundwater interaction such as the one below:

- Shimp, J. F., J. C. Tracy, L. C. Davis, E. Lee, W. Huang, L. E. Erickson, and J. L. Schnoor. Beneficial effects of plants in the remediation of soil and groundwater contaminated with organic materials. Critical Reviews in Environmental Science and Technology 23, 1, 41-77

- Line 118. Add the principal aim and the specific objectives to end your introduction.

Material and Methods

Lines 128-131. That’s clear to a reader that you deal with both surface and groundwater. However, you need to provide more details on the “geometry” of your experiment on (i) thickness of the soil zone, (ii) depth interval of the boreholes, and (iii) sampling techniques for the groundwater. Did you use a bailer or a purging technique?

Lines 128-131. Do you sample the un-saturated or saturated aquifer zone?

Lines 128-131. You need to specify rock-type and age of the aquifer that you’re studying. See also my comment later in your discussion

Line 155. “These included” you need to repeat the world analysis/analyses after “these”.

Results

Lines 174-175. “Empirical…to pooled data” unclear also at second guess. Add more detail and consider the possibility to use a synonym for “polled”. Note that, the sentence also needs more words

Line 184. Fluctuations in DOC and high values. Possible to interpret something more on biological activity in the soil from that?

Discussion

Lines 343-352. The discussion is very long. Consider applying a 10-20% reduction and make sure that you highlight the “take-home” message of the paper

Line 373. You say that the geological characteristics of the region are important, but you don’t provide any detail. Please, provide before more detail of the type of aquifer (type of rock and age)

Line 374. The same for the climate, make sure that you specify the type of climate, humid?

Conclusions

Lines 535-555. You need to insert sentences at the beginning and at the end of the bulletin points. You cannot use only bulletin points

Lines 435-555. Insert in a conclusive remark after the bulletin points the “take-home” message of the paper. Note that, all the points appear having the same importance and scientific impact on the audience in that way

References

Line 375. Insert at least the two relevant papers above

Figures and tables

Figures 2-4. Consider the use of letters (a, b, c, d…) to distinguish the different plots

Figures 2-4. Consider inserting a legend to explain symbols and the different colours. The captions are not exhaustive

Figure 5. Move closer to each other the two grids (I and II). In that way, you can make them larger and easier to read

Figure 7. Make the figure larger and improve the graphic resolution (more pixels)

Author Response

Dear Reviewer,

As a team of authors, we would like to thank you for the thorough analysis of our article and the attached comments, additions and corrections, which will certainly improve our scientific workshop and the possibility of publishing scientific papers. We tried to address all the detailed comments we found in the review. We would like to add that some of the comments relate to the lack of precision in the translation and excessive trust for the Native Speaker, who, as it turned out, was pushing the wrong forms of translation. Therefore, below we present references to individual detailed comments from the review:

Line 18 - The issue of improving the translation by the Native Speaker in the original Polish were hydrological conditions.

Lines 51-52 – Although, our research did not concern the areas directly exposed to the impact of intensified agriculture (the areas of the Polesie National Park, its buffer zone and its immediate vicinity), we decided to insert a careful annotation regarding the impact of agriculture on the quality parameters of groundwater (DOC and nitrates) as suggested.

Lines 120-171 - We do not really understand how to logically integrate information on the impact of farm animals on the environment with information on the methodology of research. Probably a mistake in the line numbering. Therefore, we decided to add an appropriate annotation to the previous supplement. We still mean a small possibility of agricultural impact on the location of research sites, hence the information is perfunctory.

Line 118 - We have supplemented the Introduction with the main goal and additional goals as suggested. Previously, we set a default target resulting from the context of the statement.

Lines 128-131 - We have supplemented the information with the required issues. In the original version of our manuscript they were described in more detail, but when translating into English, we decided to use significant abbreviations to adapt to the requirements of the Journal. In addition, we thought that the mention of the application of the ecological field research procedure developed and published by us in the peat bogs of Eastern Poland, verified by our many years of work in these areas, would be sufficient information. The question of the geological analysis of the research sites raises some interpretational doubts. With the depth of water abstraction from the peat seam (shallow groundwater) reaching 90 cm, information on the geological cross-section for different geological periods does not seem necessary. There is no water supply from deep aquifers here. Atmospheric and surface feed and infiltration of subcutaneous waters occur, as is the case with typical peat bogs. However, we tried to include this information in a common sense.

Line 155 - Changed as suggested.

Lines 174-175 - We've corrected this sentence a bit. We hope it will be more readable. After consulting our statistician, we must conclude that the term "pooled" has a strict statistical meaning and should not be replaced with any other synonym.

Line 184 - Fluctuations in the values of DOC and other physical and chemical factors are related to the internal metabolism of the fen, which is discussed in more detail in the Discussion. We do not believe that the Results section is an appropriate place for this type of deliberation. In our opinion, the principle here is a clear presentation of research data.

Lines 343-352 - We agree that the Discussion is very long. However, this is due to a wide spectrum of research in relation to the specificity of the region's habitats. To facilitate orientation for the reader, sub-chapters related to specific research topics were used. As suggested, we added information on the geology, soil, vegetation and climate of the region, which, however, again added a little bit of coverage to this chapter.

Lines 535-555 - We have added introductory and final sentences to the Conclusions, although we have not met with such an obligatory necessity so far. Similarly, in our opinion, additional explanations in the conclusions unnecessarily increase the volume of work. We tried to make the conclusions underlining the research results, be legible and easy to understand. Therefore, it seems to us that they are part of the "take-home" message.

Line 375 (? Numbering error) - In the References section we have inserted some additional publications, including the suggested ones.

Fig. 2-4, 5, 7 - We would like to clarify that the graphic materials are sent to the editorial office in separate files, and their quality is the highest possible. As authors, we have no influence on the size, resolution and layout in which the editors will place our materials in the article format. It is worth adding that this issue also applies to each separate diagram in the following figures. When creating box plots, we used standard descriptions of various elements indicating specific statistical data. This has its reference in the captions under these materials. We would like to add that which formula is so common that it should not raise any doubts.

Taking into account the attached corrections and additions, we sincerely hope that they will satisfy the Reviewer and allow our article to be published in the Water Journal.

Yours faithfully,

On behalf of the team of co-authors                                                                     

Reviewer 2 Report

Title: The importance of groundwater quality and other habitat parameters for effective active protection of an endangered plant species in Eastern Poland

The groundwater quality and biocoenotic conditions may cause notable impacts on the surrounding plants. This study attempted to explore the functions of hydrochemical and biocoenotic conditions on an endangered plant species Salix lapponum. The authors investigated water quality parameters (nitrogen, phosphorus, dissolved organic carbon, electrical conductivity and pH), S. lapponum populations, colony forming units and species of bacteria and fungi. The sampling and laboratory analysis have been performed in the appropriate way. However, the manuscript needs to be well constructed, and writing skills should be improved. My major concerns are: 1) It is required to include the consistent logic throughout the paper (particularly in Result and Discussion Sections); 2) groundwater quality and habitat parameters as collective key words in the title, their links (why you chose these objects in your research) should be clarified. Overall, this manuscript needs a solid round of improvements, and the detailed comments are as follows:

General comments:

  1. Introduction Section: Please detail your objectives or elaborate on your scientific hypothesis about the research in the section of “Introduction”.

  1. Result Section: The numerical values of parameters should be detailed in the section of “Result” for comparison and verification by other studies.

  1. Discussion Section: In this section, you should essentially discuss scientific problems based on your data from the “Result” section. There were excessive descriptions about previous conclusion and inferences without data basis, which weaken your own contributions.

Specific comments:

  1. L29, Elaborate on what “the highest hydrochemical stability” in this study refers to

  1. L36, I don't see you investigated “hydrographic condition” in this manuscript. Is that “hydrochemical”?

  1. L102, Please introduce regional meteorological, hydrological, geological and vegetation conditions in the study area.

  1. L145 and L427, General abbreviations: Dissolved organic carbon (DOC); Total organic carbon (TOC); Dissolved organic matter (DOM).

  1. L183-185, Please mark “p < 0.05”, “p < 0.01” for significant statistical differences. Please check throughout the manuscript.

  1. L228, Correlations are from “Spearman” or “Pearson”?

  1. L372-376, There were no data basis for geological, topographic, climate conditions, water management and land use that discussedin L372-376. Similar concerns could be found elsewhere in the manuscript. I suggest you focus on your data and highlight your own contributions.

Author Response

Dear Reviewer,

As a team of authors, we would like to thank you for the thorough analysis of our article and the attached comments, additions and corrections, which will certainly improve our scientific workshop and the possibility of publishing scientific papers. We tried to address all the detailed comments we found in the Review. We would like to add that some of the comments relate to the lack of precision in the translation and excessive trust for the Native Speaker, who, as it turned out, was pushing the wrong forms of translation. Therefore, below we present references to individual detailed comments from the review:

  1. We are somewhat surprised by the statement that there is no consistent logic throughout the document, in particular in the Results and Discussion chapters. Both the undertaking of the research and the method of their presentation and scientific verification result from the previously published procedure of research and analysis methods for peat bogs. We would like to explain that the chapter Results consistently contains data from the reference to the research methods: hydrochemistry with statistical analyses and biocenotic conditions with statistical analyses. This is referenced in the Discussion chapter, where two relevant subsections with relevant content are similarly distinguished. In our opinion, this is a logical consequence of the presented manuscript structure. Similarly, the contents of the Results section are based strictly on the presentation of the collected data, so it is difficult to lack logic in this aspect. A separate issue is the chapter Discussion, which requires the necessary corrections and additions, in particular regarding the geology, climate and nature of the region's vegetation. Firstly, we supplemented the Methods section, where we presented the geological background and its lack of influence on the uptake of shallow groundwater in the fen, and supplemented the Discussion section with the required information.
  2. The content of our manuscript, which is supported by many years of scientific practice and many source materials, shows that shallow groundwater in a peatland has a significant impact on biocoenoses of peatland ecosystems. Peat bogs are supplied with water from the atmosphere, both on the surface and subsurface by infiltration of rainwater or watercourses, hence the hydrochemical specificity of shallow groundwater in a decisive way shapes the hydro-dependent biocenoses. Therefore, it influences the habitat conditions for rare and protected plants. We have been successfully carrying out similar research for many species of protected and herbal plants for several dozen years. The combination of these two terms is, in our opinion, obvious and therefore fully justified.

General comments:

Introduction section - We supplemented the Introduction with the main goal and additional goals as suggested. Previously, we set a default target resulting from the context of the statement.

Results section - We took into account the information on the average numerical values of the parameters mentioned in the text as suggested. In our opinion, a wider presentation of numerical results (ranges of values, median, etc.) for individual physico-chemical factors in connection with their graphic presentation would be an unjustified duplication of information, not used in scientific publications.

Discussion section - We have completed information on the geology, climate and vegetation of the region. We agree with the opinion that the discussion is very long. This, however, is due to the wide spectrum of research carried out and the specific nature of the habitats of this region. The use of sub-chapters related to the research undertaken is helpful here. In our opinion, referring only to the research results in the Discussion is an unjustified simplification. The regional context of the issue is then lost, which is extremely important in understanding the phenomena and processes related to the internal metabolism of the fen. Therefore, we ask you to understand and accept our point of view.

Specific comments:

L29 - We would like to kindly inform you that "the highest hydrochemical stability" in this context means maintaining the values of physico-chemical parameters at a relatively unchanged level (adequate for the development needs of the species under study) during both research periods. In this case, it concerned the location of the natural occurrence of Salix lapponum (B - Bikcze).

L36 - Unfortunately, it is a matter of Native Speaker translation improvement. There were hydrological conditions in the original Polish, but in the end we decided to change this term to hydrochemical conditions, which corresponds more to the nature of our work.

L102 - We would like to kindly inform you that, referring also to the second Review, we have decided to complete this information in the appropriate places in the Methods and Discussion section.

L145 and L427 - We corrected the incorrect and variable translation of the Native Speaker. Of course, it is about Dissolved Organic Carbon (DOC).

L183-185 - We would like to inform you that here we describe the relationships between the mean values without using the statistical test, so p-values are not included. The test was used in further considerations when dividing the data into 3 locations B, D, M - which was included in line 211 (254 new version) - p <0.05.

L228 - In this case, we used Pearson to analyse the correlation of values between the physical and chemical factors.

L372-376 - For the reasons outlined above, we have decided to complete the suggested content in the Discussion. We have not studied most of these issues in the context of direct and indirect human impacts on the natural environment of the region, although, in our opinion, it has an extremely important cognitive significance. The regional context is also required in another review, which influenced our decision on this matter.

Taking into account our comments, the attached corrections and additions, we sincerely hope that they will satisfy the Reviewer and will allow our article to be published in the Water Journal.

Yours faithfully,

On behalf of the team of co-authors                                                                                                     

Round 2

Reviewer 2 Report

Dear editor:

   It is suggested that the paper will be accepted.